# The Pan-Cancer Multi-Omics Landscape of FOXO Family Relevant to Clinical Outcome and Drug Resistance

**DOI:** 10.3390/ijms232415647

**Published:** 2022-12-09

**Authors:** Jindong Xie, Junsheng Zhang, Wenwen Tian, Yutian Zou, Yuhui Tang, Shaoquan Zheng, Chau-Wei Wong, Xinpei Deng, Song Wu, Junxin Chen, Yunxian Mo, Xiaoming Xie

**Affiliations:** 1State Key Laboratory of Oncology in South China, Sun Yat-sen University Cancer Center, Collaborative Innovation Center for Cancer Medicine, Guangzhou 510060, China; 2Breast Disease Center, The First Affiliated Hospital of Sun Yat-sen University, Guangzhou 510060, China; 3Department of Endocrinology and Diabetes Center, The First Affiliated Hospital of Sun Yat-sen University, Guangzhou 510060, China

**Keywords:** pan-cancer, FOXO family, tumor microenvironment, multi-omics, clinical outcome, drug resistance

## Abstract

The forkhead box O (FOXO) transcription factors (TFs) family are frequently mutated, deleted, or amplified in various human cancers, making them attractive candidates for therapy. However, their roles in pan-cancer remain unclear. Here, we evaluated the expression, prognostic value, mutation, methylation, and clinical features of four FOXO family genes (FOXO1, FOXO3, FOXO4, and FOXO6) in 33 types of cancers based on the Cancer Genome Atlas (TCGA) and Genotype Tissue Expression (GTEx) databases. We used a single sample gene set enrichment analysis (ssGSEA) algorithm to establish a novel index called “FOXOs score”. Moreover, we investigated the association between the FOXOs score and tumor microenvironment (TME), the responses to multiple treatments, along with drug resistance. We found that the FOXO family genes participated in tumor progression and were related to the prognosis in various types of cancer. We calculated the FOXOs score and found that it was significantly correlated with multiple malignant pathways in pan-cancer, including Wnt/beta-catenin signaling, TGF-beta signaling, and hedgehog signaling. In addition, the FOXOs score was also associated with multiple immune-related characteristics. Furthermore, the FOXOs score was sensitive for predicting the efficacy of diverse treatments in multiple cancers, especially immunotherapy. In conclusion, FOXO family genes were vital in pan-cancer and were strongly correlated with the TME. A high FOXOs score indicated an excellent immune-activated TME and sensitivity to multiple treatments. Hence, the FOXOs score might potentially be used as a biomarker in patients with a tumor.

## 1. Introduction

Forkhead box (FOX) proteins are a vast family of transcription factors (TFs) known for their winged-helix DNA binding domain [1,2,3]. FOX families can regulate various biological processes, including the metabolism, proliferation, invasion, migration, and longevity [4]. Additionally, some subgroups of FOX proteins have been found to be correlated with tumor progression and metastasis [5,6]. For instance, the overexpression of the FOXM1 gene has been detected in multiple cancer types, which reveals its oncogenic potential [6,7,8]. Pan-cancer analysis showed that the FOXM1 expression in cancer was related to genomic instability and poor prognosis [5,9]. However, FOXO proteins, the class O subgroup of the FOX family, are commonly considered as tumor suppressors by inhibiting the cell cycle, promoting cell death, as well as enhancing the stress resistance [3]. In humans, the FOXO family has four members, including FOXO1, FOXO3, FOXO4, and FOXO6 [3,10]. They have been reported to be associated with tumor development, invasion, and metastasis [2,3,11]. They can lead to genetic diseases, metabolic disorders, the deregulation of aging, and cancers when dysregulated, misexpressed, and/or mutated.

Numerous studies have demonstrated a relationship between FOXO genes expression and tumor progression and metastasis [12,13,14,15]. According to a recent study, FOXO1 silencing can decrease epithelial-mesenchymal transitions (EMT) in hepatocellular carcinoma [16]. In addition, there is also strong evidence that FOXO3 is involved in the metastasis of multiple cancers, including the breast, kidney, and pancreatic cancers [17,18,19]. Similarly, the overexpression of FOXO6 inhibits the invasiveness and migration of breast cancer cells [20]. Despite these findings, other studies have linked the high expression of FOXO genes to the enhanced metastasis and poor prognosis of patients [21,22,23]. For example, a high expression level of FOXO1 and FOXO3 has been found to be related to cancer metastasis and a matrix metalloproteinase (MMP) upregulation [24,25]. The overexpression of the FOXO1 gene can also induce podocyte EMT in high glucose conditions [26]. Additionally, previous studies have demonstrated that FOXO3 promoted the invasion and migration of tumor cells by inducing an MMP expression [27,28]. These findings suggest that FOXO genes might play cell type-specific roles in tumor migration and metastasis. Therefore, the relationship and underlying mechanisms between the FOXO family and specific types of cancer need a further elucidation.

Our study firstly analyzed the FOXO family genes in pan-cancer, including the expression, prognostic value, mutation, methylation, and clinical features. The tumor microenvironment (TME), serving as the soil for seeds (cancer cells), has a crucial role in tumor development and metastasis [29,30]. In addition, the immune-activated TME, established by immune effector cells that can inhibit tumor growth, is tightly related to various treatments, including chemotherapy, radiotherapy, targeted therapies, and immunotherapy [31,32,33,34,35]. Therefore, a FOXOs score was calculated, and its association with the TME, diverse treatment responses, and drug resistance was also explored.

## 2. Results

### 2.1. Expression Level of FOXO Family Genes

Based on the Cancer Genome Atlas (TCGA) and Genotype-Tissue Expression (GTEx) database, we first investigated the expression level of four FOXO family genes in pan-cancer. They were frequently differentially expressed in 28 tumor types, as shown in Figure 1A. Uterine corpus endometrial carcinoma (UCEC) exhibits the consistently decreased expression of four FOXO family genes, whereas lower grade glioma (LGG) and pancreatic adenocarcinoma (PAAD) display a consistently increased expression. Following that, we investigated the expression of each FOXO family gene between the tumor and adjacent normal tissues in various tumor types (Figure 1B). A significant upregulation of the FOXO family genes was found in the adjacent normal tissues in pan-cancer, indicating that they might be tumor suppressors in most tumor types.

### 2.2. Prognostic Value and Single Nucleotide Variation (SNV) of FOXO Family Genes

We performed Kaplan–Meier (K–M) analysis of each FOXO gene in pan-cancer. We found that FOXO3 was a significantly protective factor in disease-specific survival (DSS), overall survival (OS), and progression-free survival (PFS) for kidney clear cell carcinoma (KIRC), uveal melanoma (UVM), and colon cancer (COAD) (Figure 2A). In addition, the FOXO4 was a significantly protective factor in DSS, OS, and PFS for thymoma (THYM), mesothelioma (MESO), LGG, and KIRC, as well as a high risk factor in disease-free survival (DFS), DSS, OS, and PFS for pheochromocytoma and paraganglioma (PCPG). SNV mainly refers to the variation in a DNA sequence caused by the alteration of a single nucleotide at the genomic level, which is widely involved in tumor initiation, development, and metastasis. We further explored the variant landscape of the FOXO family genes. The results were shown in Figure 2B–D. The total deleterious mutation percentage showed no mutation of the FOXO6 gene in pan-cancer. In contrast, the SNV mutation of the other three FOXO family genes frequently occurred in multiple tumors, especially in UCEC. Multiple kinds of variation were found, including frameshift deletion mutation, missense mutation, nonsense mutation, frameshift insertion mutation, inframe deletion, splice site, etc. A further finding was that FOXO1 and FOXO4 were the most frequently mutated (39%) among FOXO family genes in pan-cancer, and FOXO3 also had a high mutation frequency (34%).

### 2.3. The Copy Number Variation (CNV) and the Methylation Levels of FOXO Family Genes

Except for SNV, CNV, and DNA, methylation is also associated with tumor progression. It is well known that abnormal CNV is one of the critical molecular mechanisms of tumor development. We further analyzed the CNVs of each FOXO family gene. The CNV patterns in pan-cancer (including homozygous amplification, heterozygous amplification, homozygous deletion, and heterozygous deletion) are shown in Figure 3A. We found that the CNV profile of the FOXO family was mainly heterozygous, and the heterozygous deletion was more frequent than the heterozygous amplification (Figure 3B). A total of 31 of 33 tumor types showed a positive correlation between the CNV of FOXO3 and its mRNA level (Figure 3C). DNA methylation is an epigenetic modification that leads to tumorigenesis and cancer progression by silencing tumor suppressor genes. We found that the methylation levels of the FOXO family genes are increased in most tumor tissues, indicating that they generally act as tumor suppressors (Figure 3D). Besides, Figure 3E showed that the expression of the FOXO family genes was negatively associated with their promoter methylation levels in pan-cancer, especially in THYM.

### 2.4. Establishment of FOXOs Score and the Correlations between FOXOs Score and Hallmark Pathways

To calculate the FOXOs score, we used the single-sample gene set enrichment analysis (ssGSEA) algorithm in the TCGA cohort. The result showed that the distribution of the FOXOs score in each cancer type is roughly uniform, and UVM has the highest expression level of the FOXOs score (Figure 4A). A further analysis of the FOXOs score was conducted by calculating the scores of 50 hallmark pathways in pan-cancer pathways. We found that the FOXOs score was strongly correlated with these pathways (Appendix A). We then investigated the relationship between the FOXOs score and the scores of 50 hallmark pathways in each cancer type (Figure 4B). According to our findings, the FOXOs score was positively related to many classic pathways, including Wnt/beta-catenin signaling, TGF-beta signaling, and hedgehog signaling, while negatively associated with DNA repair and MYC signaling, which indicated a poor immune response.

### 2.5. The Correlations between FOXOs Score and TME as well as Immune Checkpoints

A further analysis was conducted to determine whether there was a correlation between the FOXOs score and TME in pan-cancer. It was evident from the heatmap that most immune cells were upregulated as the FOXOs score increased, which was confirmed by different algorithms (Figure 5A). The result was then found consistent in each cancer type (Appendix A). In addition, we found that the FOXOs score was positively related to the expression of classic immune checkpoints such as CD28, CTLA4, etc. (Figure 5B)

### 2.6. The Correlations between FOXOs Score and Stemness and Immunogenicity

Stemness and immunogenicity were found to be connected with tumor progression. By investigating the correlation between the FOXOs score and them, a negative correlation was found between the FOXOs score and the RNA stemness score (RNAss) in pan-cancer. However, other indicators such as the tumor mutation burden (TMB), DNA stemness score (DNAss), and microsatellite instability (MSI) varied in different types of cancer (Figure 6A).

### 2.7. The Predictive Efficacy of FOXOs Score in Multi-Type of Cohorts

We firstly collected several immunotherapy cohorts (GSE91061, CheckMat, and GSE78220) and calculated the FOXOs score in each cohort. The K–M analysis revealed that patients receiving immunotherapy with a high FOXOs score had an improved OS or PFS (Figure 6B). We assessed the predictive efficacy using the receiver operating characteristic (ROC) curve analyses based on the FOXOs score. The result showed that the area under the curve (AUC) value reached 70.7% (95%CI: 45.9–95.6%), 59.0% (95%CI: 48.6–69.3%), and 70.7% (95%CI: 35.1–82.9%) in GSE91061, CheckMat, and GSE78220 cohorts, respectively (Figure 6C). Higher FOXOs scores led to more responsive patients in each cohort, indicating that patients with a high FOXOs score might benefit from immunotherapy (Figure 6D).

We further evaluated whether the FOXOs score could predict the survival outcomes of patients with post-surgery or chemotherapy in diverse cohorts. We found that post-surgery patients with a high FOXOs score had a better OS, recurrence-free survival (RFS), and PFS (Figure 7A). Meanwhile, patients undergoing chemotherapy with a high FOXOs score also had a better OS, RFS, PFS, and DSS (Figure 7B). By further analyzing the effect of the FOXOs score and chemotherapy on the survival outcomes, we found that patients undergoing chemotherapy with a high FOXOs score had the best OS. In contrast, those who did not receive chemotherapy with a low FOXOs score had the worst OS (Figure 7C).

### 2.8. The Correlations between FOXOs Score and Drug Sensitivity, and Single-Cell RNA (scRNA) Transcriptome Analysis of the Distribution of FOXOs Score

To further understand whether the FOXOs score could predict the patients’ response to diverse therapies, we downloaded the expression and drug sensitivity data in cancer cell lines from the Genomics of Drug Sensitivity in Cancer (GDSC) database. We examined the relationship between the FOXOs score and the half-maximal inhibitory concentration (IC50) values of the drugs. Among pan-cancer, especially in breast cancer (BRCA), we found significant negative correlations between the FOXOs score and IC50 values, which suggests that a high FOXOs score might increase the sensitivity to chemotherapy and targeted drugs (Figure 8A). To verify the consistency of the conclusions obtained with bulk transcriptome data, we collected scRNA transcriptome data (GSE176078, GSE149614, and GSE160269) to analyze and visualize the enrichment scores of the FOXOs score in malignant cells as well as normal epithelial cells. As we thought, the FOXOs score in malignant cells was found to be lower than that in adjacent normal epithelial cells among these datasets, which confirmed our findings (Figure 8B,C).

## 3. Discussion

TFs, the ultimate effector molecules of cellular function, are critical regulators of cell division and death [36]. Many pathological conditions are associated with the dysregulation of TFs. The FOXO family genes, known for their winged-helix DNA binding domain, generally suppress tumor growth by inhibiting the cell cycle, promoting cell death, and enhancing the stress resistance [37,38]. In addition, scientists have found that mutations, deletions, or amplifications of the FOXO family genes existed in various human cancers, indicating that the FOXO family genes might be attractive therapeutic targets. However, their roles in pan-cancer remain unclear, and the potential antitumor mechanisms need a further elucidation.

Based on the TCGA and GTEx databases, we examined the expression and prognostic value of four FOXO family genes in pan-cancer. The result showed that the FOXO family genes, especially FOXO1 and FOXO4, were lower expressed in tumor tissues in most tumors, consistent with the knowledge that the FOXOs are considered to be tumor suppressors, and the dysregulation of the FOXOs can cause diverse pathological conditions, including cancer. Then, we conducted a comprehensive analysis of genetic alterations in FOXO family genes, including mutations and methylations. The result showed that the deletion of FOXO1 and FOXO4 genes, and the increase in CNVs in the FOXO3 gene, were linked to tumor progression. The overexpression of FOXO6 has previously been demonstrated to inhibit the migration and invasion of breast cancer cells [20]. Moreover, FOXO triple-knockout mice (FOXO1/3/4−/−) were prone to developing hemangiomas (endothelial cell hamartomas) and thymic lymphomas, confirming the FOXO proteins as genuine tumor suppressors [39]. A previous study showed that silencing FOXO1 in hepatocellular carcinoma can enhance EMT [16]. In addition, there is strong evidence that the FOXO3 is involved in the metastasis of multiple cancers, including breast, pancreatic, and kidney cancers [17,18,19]. Despite overwhelming evidence supporting the FOXO family genes as tumor suppressors, some experts argued that the FOXO family genes could also be oncogenic. Previous studies have shown that the high expression levels of FOXO genes are correlated with a poor prognosis and enhanced metastasis of patients [21,22,23]. For example, a high expression level of FOXO1 and FOXO3 has been found to be correlated with cancer metastasis and an MMP upregulation [24,25]. Our study found that the FOXO4 was a significantly protective factor in DSS, OS, and PFS for THYM, MESO, LGG, and KIRC, while a high risk factor in DFS, OS, and PFS for DLBC (diffuse large B-cell lymphoma). The previous study consistently revealed that diffuse large B-cell lymphoma resistant to treatment exhibited an enhanced FOXO4 expression and stem cell-like properties, reflecting a significant association between the expression of FOXO4 in DLBC and a poor prognosis [23]. Collectively, these results suggest that the FOXO family genes are commonly differentially expressed in pan-cancer and have different prognostic values in various cancers. However, the underlying mechanisms need further elucidation.

The TME has been found to play a vital role in tumor development and metastasis [30,40,41,42]. In particular, the TME containing immune effector cells can be used to test the efficacy of immunotherapy [31,32,33,43,44]. Based on the ssGSEA algorithm, we calculated the FOXOs score. We found that it was significantly correlated with multiple malignant pathways in pan-cancer, such as Wnt/beta-catenin signaling, TGF-beta signaling, and hedgehog signaling, while negatively associated with DNA repair and MYC signaling, which indicated a poor immune response, and all of which were closely related to the TME. FOXO proteins are critically required to maintain the survival of naive T cells and to coordinate the differentiation of effector and memory T cells [45]. For example, the FOXO1 can modulate the migration of T cells by enhancing the expression of lymphoid organ-homing molecules [46]. Additionally, the manipulation of FOXO1 can regulate the PD-1 axis in the tumor-infiltrating T cells, which are the primary immune effector cells of the TME, indicating a novel therapeutic strategy for cancer treatment [45]. Therefore, we examined the relationship between the FOXOs score and TME as well as immune checkpoints. Consistent with previous studies, our results showed that as the FOXOs score increased, most immune cells were upregulated, which was confirmed by different algorithms. Furthermore, we found that the FOXOs score was positively related to the expression of classic immune checkpoints such as CD28, CTLA4, etc.

In addition, we found that patients receiving immunotherapy with a high FOXOs score had improved OS or PFS, and higher FOXOs scores led to more responsive patients in each cohort, indicating that patients with a high FOXOs score might benefit from immunotherapy. Our studies also showed that the patients undergoing surgery or chemotherapy with a high FOXOs score had a better OS, RFS, and PFS in multiple cohorts which is consistent with previous studies [47,48,49,50]. We then assessed whether the FOXOs score could reflect the response of patients in diverse treatments, and the results were satisfactory. Moreover, we analyzed the correlation of the FOXOs score with multiple drugs. We found that the IC50 values of most drugs were negatively correlated with the FOXOs score in pan-cancer, especially BRCA, confirming that a high FOXOs score might increase the sensitivity to chemotherapy and targeted drugs. Collectively, these findings disclose that the FOXOs score strongly correlates with the TME and might be a potential biomarker to predict the efficacy of multiple treatments, especially immunotherapy.

## 4. Materials and Methods

### 4.1. Data Collection

We obtained normalized and log2 converted RNA-sequencing (RNA-seq) profiles transcripts per million (TPM) and the corresponding clinical information of the TCGA and GTEx from the UCSC Xena database. To convert the ensemble ids to the gene symbols, the “GeoTcgaData” R package was used. Multiple therapeutic cohorts were downloaded from the Gene Expression Omnibus (GEO) database (ID: GSE91061, GSE78220, GSE31210, GSE62452, GSE1456, GSE39582, GSE72970, GSE169455, GSE48277, GSE14814, GSE14208, and GSE96058) and a previous study (CheckMat) [33,51]. In case of need, the probes were mapped using the “AnnoProbe” R package. Averaging multiple probes were calculated using the “limma” R package when necessary [52]. The Gene Set Cancer Analysis (GSCA) database was used to assess gene alterations in the FOXO family, including SNV, CNV, and methylation. A total of 50 hallmark pathways were obtained from the Molecular Signature Database (MSigDB) and were analyzed as previously described [53,54].

### 4.2. FOXOs Score Analysis

We calculated FOXOs score of each patient using the ssGSEA algorithm (“GSVA” R package) [55]. The formula was as follows (for a given signature *G* of size *N*_*G* and single sample *S*, of the data set of *N* genes, the genes are replaced by their ranks according to their absolute expression from high to low: *L* = {*r*_1, *r*_2, …, *r*_*N*}. An *FOXOs score* (*G*, *S*) is obtained by a sum (integration) of the difference between the weighted Empirical Cumulative Distribution Functions (ECDF) of the genes in the signature $P_{G}^{W}$ and the ECDF of the remaining genes $P_{N_{G}}$),
(1)$$FOXOs\;score\;\left(G,S\right)=\overset{N}{\underset{i=1}{\sum }}\left[P_{G}^{w}\left(G,S,i\right)-P_{N_{G}}\left(G,S,i\right)\right]$$
(2)$$where\;P_{G}^{W}\left(G,S,i\right)=\underset{r_{j}\in G,j\leq i}{\sum }\frac{{\left|r_{j}\right|}^{\alpha }}{\sum_{r_{j}\in G}{\left|r_{j}\right|}^{\alpha }}$$
(3)$$and\;P_{N_{G}}\left(G,S,i\right)=\underset{r_{j}\notin G,j\leq i}{\sum }\frac{1}{\left(N-N_{G}\right)}$$

### 4.3. Tumor Microenvironment and Immune Checkpoints Analysis

The infiltration data from the TCGA cohort on the immune cells were calculated by R package “immunedeconv” [56]. Immune checkpoints were obtained from a previous study [57].

### 4.4. Clinical Outcome Analysis

To determine whether the FOXOs score correlates with the survival outcomes (OS, DSS, DFS, and PFS), the “survival” and “survminer” R packages were used to perform a K–M analysis, as previously described [58]. We determined the best cut-point by the “surv_cutpoint” function. The “pROC” R package was used to perform the ROC curve analysis [59]. For predicting the responsive efficacy, we applied a Chi-square test and Fisher’s exact test. 

### 4.5. Drug Sensitivity Analysis

We used an “oncoPredict” R package to predict the drug’s sensitivity [60]. We obtained a training dataset and corresponding information from the GDSC as previously described [61]. The “ggplot” R package was applied to the visualization of drug sensitivity analysis.

### 4.6. Single-Cell RNA Transcriptome Analysis

We collected single-cell RNA transcriptome data from the GEO database (ID: GSE176078, GSE149614, and GSE160269) [62,63,64]. We analyzed the data above with the “Seurat V4” R package [65]. We used the “inferCNV” R package to perform the recognition of malignant and normal epithelial cells [66].

### 4.7. Statistical Analysis

All statistical analyses were presented via R 4.1.0. For analyzing the differences between the two groups, Student’s *t*-test was used. K–M plots were used to chart the survival curves and log-rank tests were performed to compare them. The correlation coefficients were calculated as Pearson correlations. The false discovery rate was calculated by the Benjamini and Hochberg method. We considered *p* < 0.05 to be statistically significant.

## 5. Conclusions

The findings of this study demonstrate that FOXO family genes are critical in tumor progression and tumorigenesis. The FOXOs score is strongly related to the TME and might serve as a biomarker for predicting the efficacy of diverse treatments, especially immunotherapy. Our study provides an insight into the potential anti-tumor mechanisms mediated by FOXO family genes, which needs further investigation and confirmation.

## Figures and Tables

**Figure 1 ijms-23-15647-f001:**
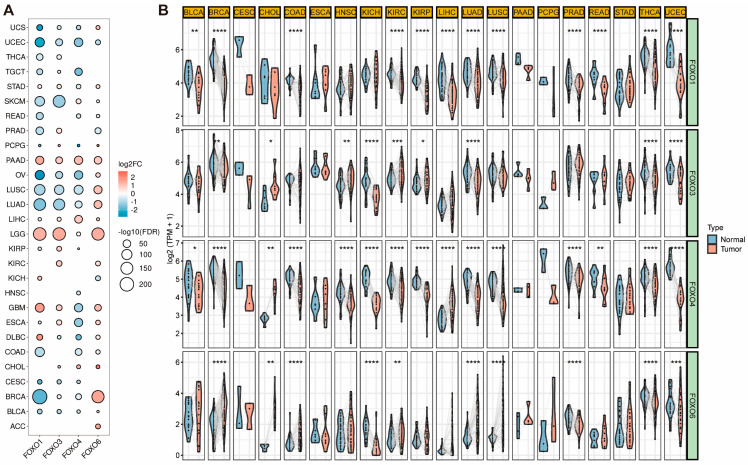
The different mRNA expression of FOXO family genes. (**A**) Bubble plot of the different mRNA expression of FOXO family between tumor and normal tissues based on the TCGA and the GTEx datasets (FC, fold change; FDR, false discovery rate). (**B**) Violin plot of the different mRNA expression of FOXO family in paired tumor and adjacent normal tissues based on the TCGA dataset (* *p* < 0.05, ** *p* < 0.01, *** *p* < 0.001, **** *p* < 0.0001). ACC, adrenocortical carcinoma; BLCA, bladder urothelial carcinoma; BRCA, breast invasive carcinoma; CESC, cervical squamous cell carcinoma and endocervical adenocarcinoma; CHOL, cholangiocarcinoma; COAD, colon adenocarcinoma; DLBC, lymphoid neoplasm diffuse large B−cell lymphoma; ESCA, esophageal carcinoma; GBM, glioblastoma multiforme; HNSC, head and neck squamous cell carcinoma; KICH, kidney chromophobe; KIRC, kidney renal clear cell carcinoma; KIRP, kidney renal papillary cell carcinoma; LGG, brain lower grade glioma; LIHC, liver hepatocellular carcinoma; LUAD, lung adenocarcinoma; LUSC, lung squamous cell carcinoma; OV, ovarian serous cystadenocarcinoma; PAAD, pancreatic adenocarcinoma; PCPG, pheochromocytoma and paraganglioma; PRAD, prostate adenocarcinoma; READ, rectum adenocarcinoma; SKCM, skin cutaneous melanoma; STAD, stomach adenocarcinoma; TGCT, testicular germ cell tumors; THCA, thyroid carcinoma; UCEC, uterine corpus endometrial carcinoma; UCS, uterine carcinosarcoma.

**Figure 2 ijms-23-15647-f002:**
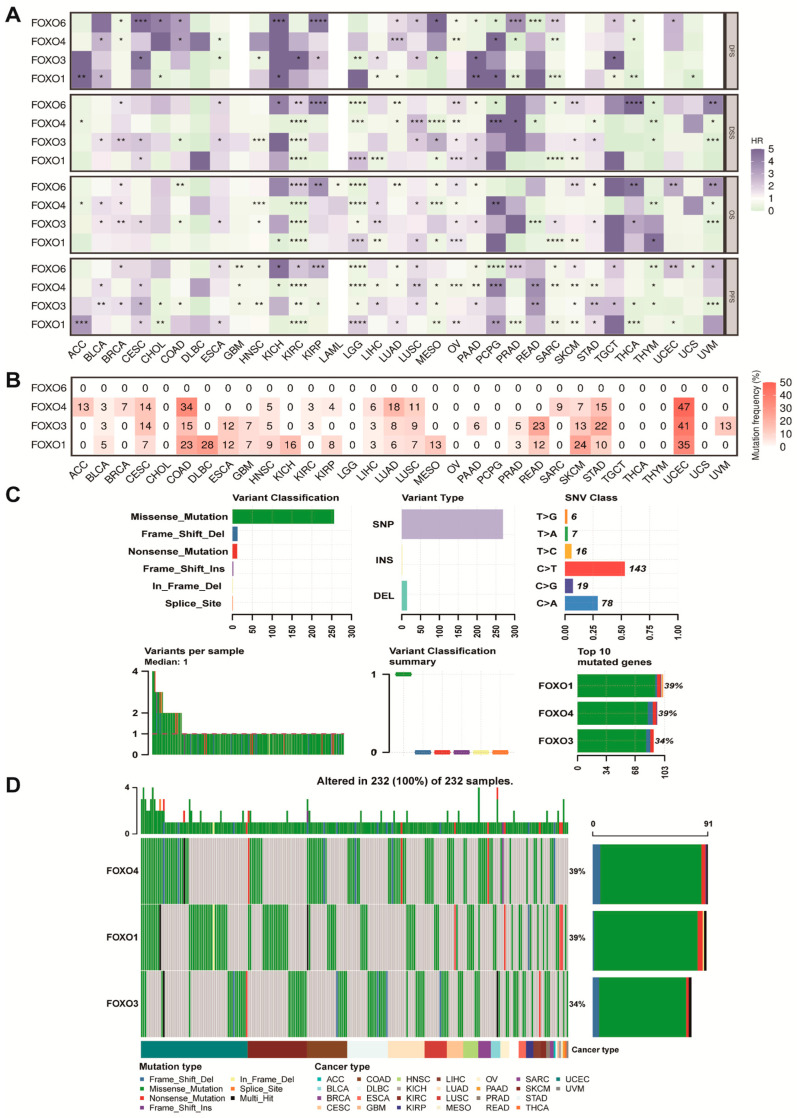
The prognostic value and the SNV alteration of FOXO family genes. (**A**) Heatmap of the prognostic value of FOXO family in each tumor type (HR, hazard ratio; * *p* < 0.05, ** *p* < 0.01, *** *p* < 0.001, **** *p* < 0.0001). (**B**) The SNV profile of FOXO family in each tumor type. (**C**) The SNV summary of FOXO family in pan-cancer. (**D**) Oncoplot of the mutation distribution of FOXO family in pan-cancer.

**Figure 3 ijms-23-15647-f003:**
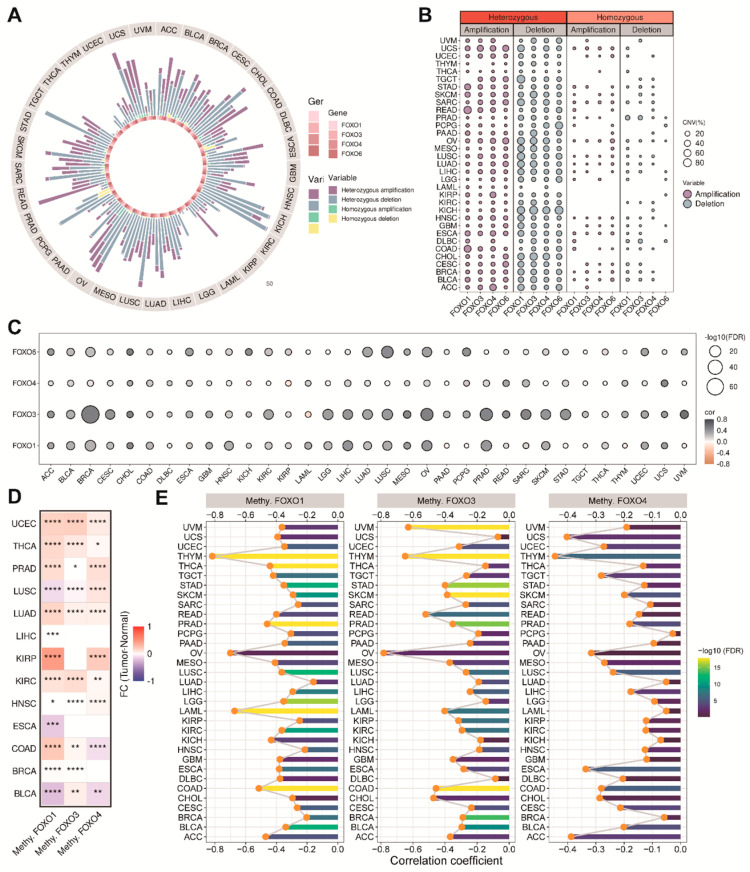
The CNV alteration and the methylation levels of FOXO family genes. (**A**) The CNV percentage of FOXO family in each tumor type. (**B**) The heterozygous and homozygous CNV profile of FOXO family in each tumor type, including the percentage of amplification and deletion. (**C**) Bubble plot of the correlations between CNV and mRNA expression of FOXO family in pan-cancer (FDR, false discovery rate). (**D**) Heatmap of the different methylation levels of FOXO family in pan−cancer (FC, fold change; * *p* < 0.05, ** *p* < 0.01, *** *p* < 0.001, **** *p* < 0.0001). (**E**) Bar plot of the correlations between the methylation levels and mRNA expression of FOXO family in pan−cancer (FDR, false discovery rate).

**Figure 4 ijms-23-15647-f004:**
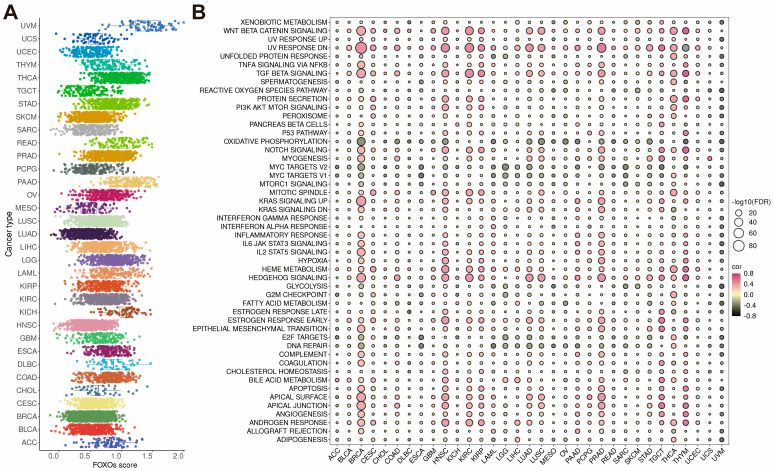
The FOXOs score distribution and the correlations between FOXOs score and hallmark pathways. (**A**) The FOXOs score distribution in the TCGA pan−cancer cohort. (**B**) Bubble plot of the correlations between FOXOs score and hallmark pathways in each tumor type. Pink indicates a positive correlation and black indicates a negative correlation. (FDR, false discovery rate).

**Figure 5 ijms-23-15647-f005:**
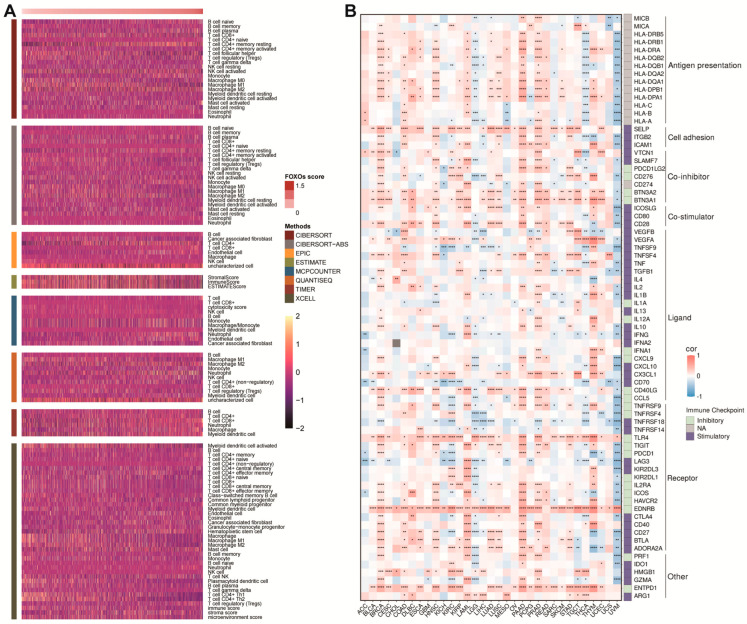
The correlations between FOXOs score and tumor microenvironment as well as immune checkpoints. (**A**) Heatmap of the tumor microenvironment scores calculated by different algorithms. (**B**) Heatmap of the correlations between FOXOs score and immune checkpoints in each tumor type (FDR, false discovery rate; * *p* < 0.05, ** *p* < 0.01, *** *p* < 0.001, **** *p* < 0.0001).

**Figure 6 ijms-23-15647-f006:**
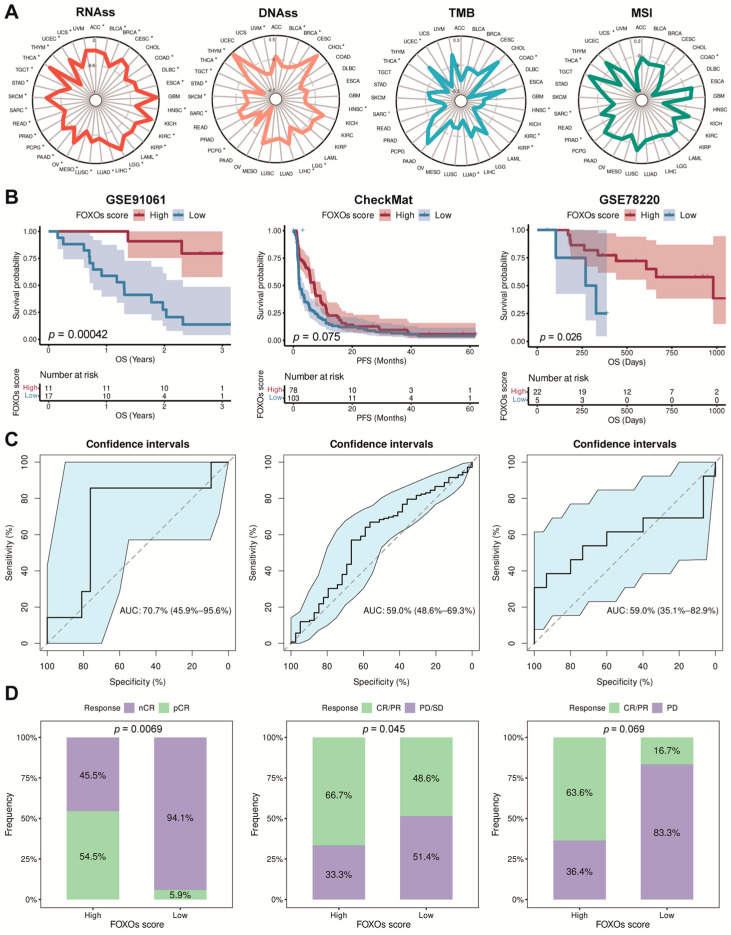
The correlations between FOXOs score and stemness and immunogenicity, as well as the predictive efficacy of FOXOs score in immunotherapy cohorts. (**A**) The correlations between FOXOs score and RNAss, DNAss, TMB, and MSI (* *p* < 0.05). (**B**) The Kaplan–Meier survival analyses of FOXOs score in GSE91061, CheckMat, and GSE78220 cohorts. (**C**) The ROC curve analyses of predicting immunotherapy efficacy based on FOXOs score in GSE91061, CheckMat, and GSE78220 cohorts. (**D**) Results of chi-square test between responsiveness and FOXOs groups in GSE91061, CheckMat, and GSE78220 cohorts.

**Figure 7 ijms-23-15647-f007:**
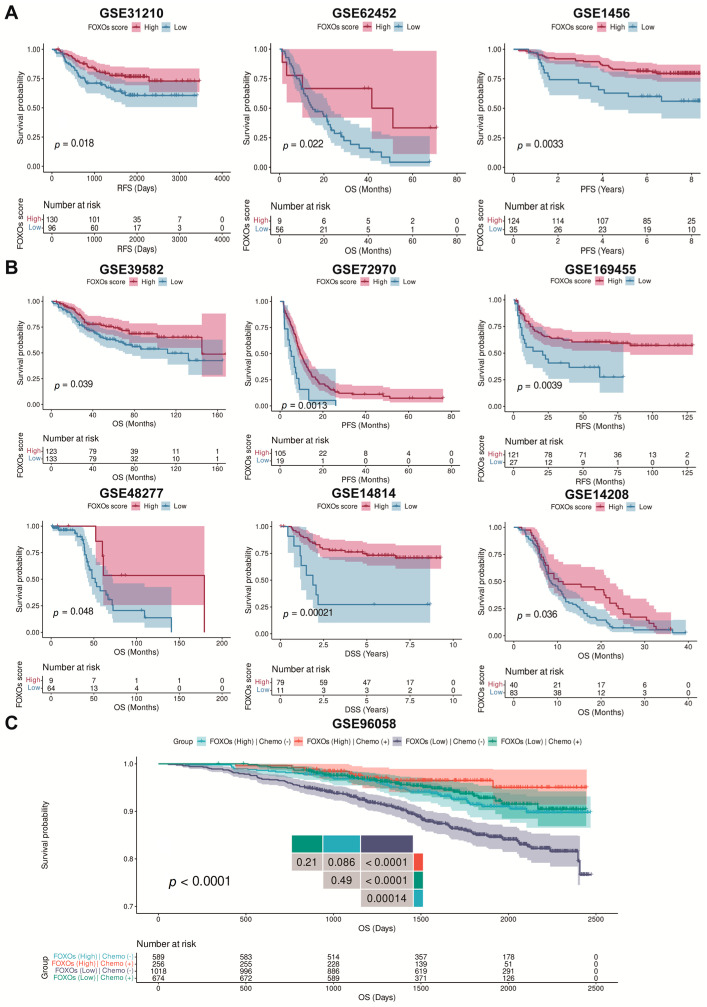
The Kaplan–Meier survival analyses of FOXOs score in multi-type of cohorts. (**A**) The Kaplan–Meier survival analyses of FOXOs score in post-surgery cohorts. (**B**) The Kaplan–Meier survival analyses of FOXOs score in chemotherapy cohorts. (**C**) The Kaplan–Meier survival analyses of FOXOs score in GSE96058 cohort.

**Figure 8 ijms-23-15647-f008:**
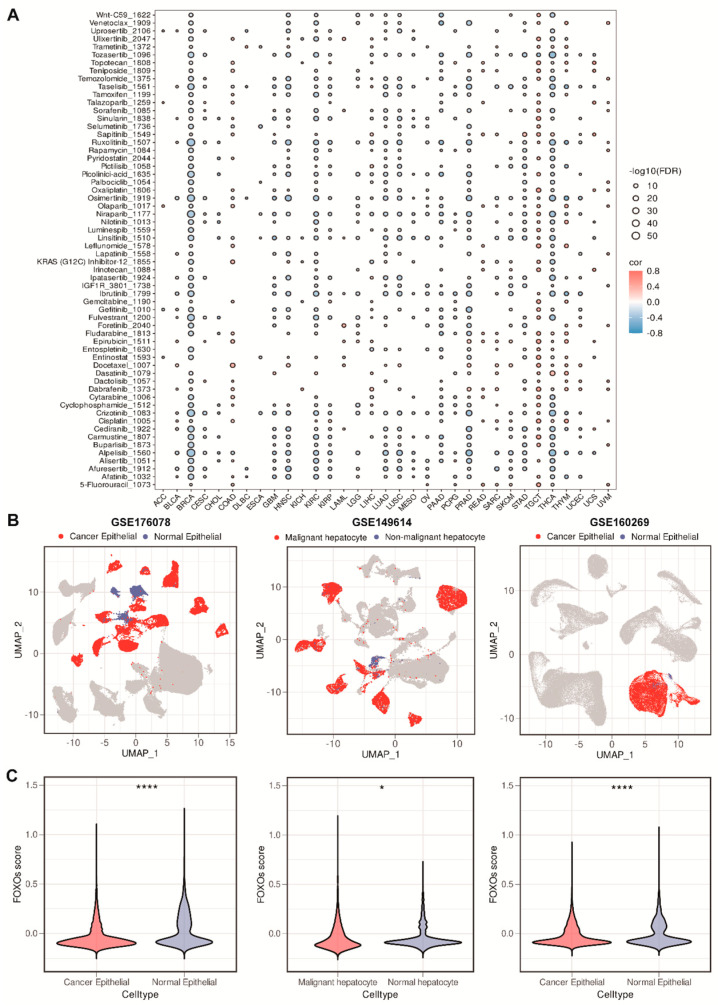
The correlations between FOXOs score and drug sensitivity and single–cell RNA transcriptome analysis of the distribution of FOXOs score. (**A**) Bubble plot of the correlations between FOXOs score and IC50 values of common drugs (FDR, false discovery rate). (**B**) UMAP plot visualization of the distribution of FOXOs score between malignant and normal epithelial cells in GSE176078, GSE149614, and GSE160269. (**C**) Violin plots of the FOXOs score between malignant and normal epithelial cells in GSE176078, GSE149614, and GSE160269 (* *p* < 0.05, and **** *p* < 0.0001).

## Data Availability

All datasets involved in this study can be viewed in the UCSC Xena database (https://xenabrowser.net/datapages/) (15 July 2022), the Gene Set Cancer Analysis (GSCA) (http://bioinfo.life.hust.edu.cn/GSCA/) (16 July 2022), the Molecular Signature Database (MSigDB) (https://www.gsea-msigdb.org/gsea/msigdb/) (18 July 2022), and Gene Expression Omnibus (GEO), or data availability part of the corresponding articles. All the other data supporting the findings of this study are available within the article and its Appendix A or from the corresponding authors upon reasonable request.

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
