# Peer review of "The Pan-Cancer Multi-Omics Landscape of FOXO Family Relevant to Clinical Outcome and Drug Resistance"

_ijms, 2022, doi:10.3390/ijms232415647_

Round 1
Reviewer 1 Report
please find attached.

Author Response
#Reviewer 1
- Abstract can be improved regarding clarity, eg “pan-cancer”, “FOXO score” might be not clear for a broad readership.
Response: Thank you for your reminding. We have improved regarding clarity of the abstract as follows,
Here, we evaluated the expression, prognostic value, mutation, methylation, and clinical features of four FOXO family genes (FOXO1, FOXO3, FOXO4, and FOXO6) in 33 types of cancers based on the Cancer Genome Atlas (TCGA) and Genotype Tissue Expression (GTEx) databases. (Page 1, Line 17-21)
We used single sample gene set enrichment analysis (ssGSEA) algorithm to establish a novel index called “FOXOs score”. (Page 1, Line 20-21)
- Introductory section is sparse and should give more background information about current knowledge.
Response: We feel sorry that we did not provide enough information in the introduction section. We have carefully added more background information based on current knowledge as follows,
Forkhead box (FOX) proteins are a vast family of transcription factors (TFs) known for their winged-helix DNA binding domain [1-3]. FOX families can regulate various biological processes, including metabolism, proliferation, invasion, migration, and longevity [4]. Additionally, some subgroups of FOX proteins have been found to be correlated with tumor progression and metastasis [5, 6]. For instance, overexpression of the FOXM1 gene has been detected in multiple cancer types, which reveals its oncogenic potential [6-8]. Pan-cancer analysis showed that FOXM1 expression in cancer was related to genomic in-stability and poor prognosis [5, 9]. However, FOXO proteins, the class O subgroup of the FOX family, are commonly considered as tumor suppressors by inhibiting the cell cycle, promoting cell death, as well as enhancing stress resistance [3]. (Page 1, Line 37-Page 2, Line 46)
Overexpression of the FOXO1 gene can also induce podocyte EMT in high glucose conditions [26]. Additionally, previous studies have demonstrated that FOXO3 promoted the invasion and migration of tumor cells by inducing MMP expression [27, 28]. These findings suggest that FOXO genes might play cell-type-specific roles in tumor migration and metastasis. (Page 2, Line 60-64)
The tumor microenvironment (TME), serving as the soil for seeds (cancer cells), has a crucial role in tumor development and metastasis [29, 30]. In addition, the immune-activated TME, established by immune effector cells that can inhibit tumor growth, is tightly related to various treatments, including chemotherapy, radiotherapy, targeted therapies, and immunotherapy [31-34]. (Page 2, Line 68-73)
- References should be revised since they do not reflect recent studies, eg cited FOXO-related studies are 8-10 years old! In general, additional references should be included, eg. concerning pan-can analysis, eg. Barger, C.J.; Branick, C.; Chee, L.; Karpf, A.R. Pan-Cancer Analyses Reveal Genomic Features of FOXM1 Overexpression in Cancer. Cancers 2019, 11, 251. https://doi.org/10.3390/cancers11020251 and others
Response: We feel sorry that we did not provide enough recent studies in our manuscript, and we agree with this suggestion. We have added relevant references in recent years as follows,
1.Yadav, R. K.; Chauhan, A. S.; Zhuang, L.; Gan, B., FoxO transcription factors in cancer metabolism. Seminars in cancer biology 2018, 50, 65-76.
2.Gong, Z.; Yu, J.; Yang, S.; Lai, P. B. S.; Chen, G. G., FOX transcription factor family in hepatocellular carcinoma. Biochimica et biophysica acta. Reviews on cancer 2020, 1874, (1), 188376.
3.Jiramongkol, Y.; Lam, E. W., FOXO transcription factor family in cancer and metastasis. Cancer metastasis reviews 2020, 39, (3), 681-709.
4.Wang, J.; Li, W.; Zhao, Y.; Kang, D.; Fu, W.; Zheng, X.; Pang, X.; Du, G., Members of FOX family could be drug targets of cancers. Pharmacology & therapeutics 2018, 181, 183-196.
5.Barger, C. J.; Branick, C.; Chee, L.; Karpf, A. R., Pan-Cancer Analyses Reveal Genomic Features of FOXM1 Overexpression in Cancer. Cancers 2019, 11, (2).
6.Barger, C. J.; Chee, L.; Albahrani, M.; Munoz-Trujillo, C.; Boghean, L.; Branick, C.; Odunsi, K.; Drapkin, R.; Zou, L.; Karpf, A. R., Co-regulation and function of FOXM1/RHNO1 bidirectional genes in cancer. eLife 2021, 10.
- Order of figures is confusing (text and figures do not fit!) and makes reading difficult. Figure 1 seems to be shrunken and is hardly readable. Legend of Fig. 1 should contain explanation of tumor type abbreviations.
Response: Thank you for pointing out this problem. We feel sorry that we did not arrange the order of figures well. We have rearranged the layout of the manuscripts. Besides, we have adjusted the scale of Figure 1. Moreover, we have added the explanation of tumor type abbreviations in Figure 1. The revision is as follows,
Figure 1. The different mRNA expression of FOXO family genes. (A) Bubble plot of the different mRNA expression of FOXO family between tumor and normal tissues based on the TCGA and the GTEx datasets (FC, fold change; FDR, false discovery rate). (B) Violin plot of the different mRNA expression of FOXO family in paired tumor and adjacent normal tissues based on the TCGA dataset (*p < 0.05, **p < 0.01, ***p < 0.001, ****p < 0.0001). ACC, Adrenocortical carcinoma; BLCA, Bladder Urothelial Carcinoma; BRCA, Breast invasive carcinoma; CESC, Cervical squamous cell carcinoma and endocervical adenocarcinoma; CHOL, Cholangiocarcinoma; COAD, Colon adenocarcinoma; DLBC, Lymphoid Neoplasm Diffuse Large B-cell Lymphoma; ESCA, Esophageal carcinoma; GBM, Glioblastoma multiforme; HNSC, Head and Neck squamous cell carcinoma; KICH, Kidney Chromophobe; KIRC, Kidney renal clear cell carcinoma; KIRP, Kidney renal papillary cell carcinoma; LGG, Brain Lower Grade Glioma; LIHC, Liver hepatocellular carcinoma; LUAD, Lung adenocarcinoma; LUSC, Lung squamous cell carcinoma; OV, Ovarian serous cystadenocarcinoma; PAAD, Pancreatic adenocarcinoma; PCPG, Pheochromocytoma and Paraganglioma; PRAD, Prostate adenocarcinoma; READ, Rectum adenocarcinoma; SKCM, Skin Cutaneous Melanoma; STAD, Stomach adenocarcinoma; TGCT, Testicular Germ Cell Tumors; THCA, Thyroid carcinoma; UCEC, Uterine Corpus Endometrial Carcinoma; UCS, Uterine Carcinosarcoma.
- FIGURE 3 is completely missing! Results remain unclear: what did authors mean by “we found that most of these methylated genes were upregulated in tumor tissues” (l.102)? Since methylated is usually associated with gene silencing and thus downregulation. This should be clarified!
Response: We are sorry if Figure 3 was lost due to the problem of uploading files. It was previously shown in Page 5. We have re-added Figure 3 and rearranged the layout of the manuscripts. As for the sentence “we found that most of these methylated genes were upregulated in tumor tissues”, we felt sorry that we made a mistake in our manuscript. We agree with your suggestion and have corrected the clarification of this sentence as follows:
DNA methylation is an epigenetic modification that leads to tumorigenesis and cancer progression by silencing tumor suppressor genes. We found that the DNA methylation levels of FOXO family genes were downregulated in most tumor tissues, indicating that they generally act as tumor suppressors (Figure 3D). Besides, Figure 3E showed that the expression of FOXO family genes was negatively associated with their promoter methylation levels in pan-cancer, especially in THYM. These findings implied that FOXO family genes might be the CNV and methylation drive genes. (Page 5, Line 143-149)
- In general, the conducted analyses are comprehensive and include a lot of data. However, most of the figures (especially, Figs 4, 5, and 8 containing bubble plots) are hardly readable and not clear to a broad readership. The authors should re-evaluate if major findings can be extracted and presented in the main figures more clearly. The rest of the data/the whole data sets might be placed in a supplementary section available for the “expert reader”.
Response: We feel great thanks for this professional suggestion. We agree with this suggestion, and we have re-evaluated the results we found, rearranged the layout of figures, and added figures related to main conclusions. We think that bubble plot is suitable to show results contained multiple information such as correlation coefficient and FDR. Therefore, we have changed the presentation of Figure 3E, put Figure 4B into Supplementary Figure S1, and put Figure 5B into Supplementary Figure S2.
- The “FOXOs score” should be shortly and simply explained to readers which are not familiar with the bioinformatic details of the ssGSEA algorithm! What does “UVM has the highest FOXOs score” actually say? Highest expression?
Response: It is a valuable suggestion. We have added the explanation of the “FOXOs score” with the formula which allows readers to read and understand the manuscript. As for the sentence “UVM has the highest FOXOs score”, what we want to express is that we give a general description about the expression level of the established FOXOs score among pan-cancer, and the distribution of the FOXOs score in each cancer type is roughly uniform. As the reviewer says, UVM has the highest expression level of the FOXOs score. We have changed the statement of this sentence and we can set Figure 4A into the supplementary figures if you think it is not important enough. The corrections are as follows,
The formula was as follows (For a given signature of size and single sample , of the data set of genes, the genes are replaced by their ranks according their absolute expression from high to low: . An is obtained by a sum (integration) of the difference between a weighted Empirical Cumulative Distribution Functions (ECDF) of the genes in the signature and the ECDF of the remaining genes ), (Page X, Line XX-XX)
The result showed that the distribution of the FOXOs score in each cancer type is roughly uniform, and UVM has the highest expression level of the FOXOs score (Figure 4A). (Page 12, Line 334-Page 13, Line342)
- Discussion can be improved regarding the classification of their results in context of current literature. See above: more recent studies can be cited! Outlook and evaluation of the consequences for the clinics is also missing.
Response: Your suggestion really means a lot to us. We have improved the discussion part and added the outlook and evaluation of the consequences for the clinics as follows,
Our study found that the FOXO4 was a significantly protective factor in DSS, OS and PFS for THYM, MESO, LGG, and KIRC, while a highly risk factor in DFS, OS, and PFS for DLBC (Diffuse Large B-cell Lymphoma). The previous study consistently revealed that diffuse large B-cell lymphoma resistant to treatment exhibited enhanced FOXO4 expression and stem cell-like properties, reflecting a significant association between FOXO4 expression in DLBC and poor prognosis [23]. Collectively, these results suggest that the FOXO family genes are commonly differentially expressed in pan-cancer and have different prognostic values in various cancers. However, the underlying mechanisms need further elucidation. (Page 11, Line 273-281)
FOXO proteins are critically required to maintain the survival of naive T cells and to coordinate the differentiation of effector and memory T cells [44]. For example, the FOXO1 can modulate T cells migration by enhancing the expression of lymphoid organ-homing molecules [45]. Additionally, manipulation of FOXO1 can regulate the PD-1 axis in the tu-mor-infiltrating T cells, which are the primary immune effector cells of TME, indicating a novel therapeutic strategy for cancer treatment [44]. Therefore, we examined the relationship between FOXOs score and TME as well as immune checkpoints. Consistent with previous studies, our results showed that as the FOXOs score increased, most immune cells were upregulated, which was confirmed by different algorithms. (Page 11, Line 288- Page 12, Line 297)
- Lack of a final “conclusion” section summarizing the most important findings.
Response: Thank you for pointing out this problem. We agree with it and have added “conclusion” section as follows,
Conclusion
The findings of this study demonstrate that FOXO family genes are critical in tumor progression and tumorigenesis. The FOXOs score is strongly related to TME and might serve as a biomarker for predicting the efficacy of diverse treatments, especially immunotherapy. Our study provides insight into potential anti-tumor mechanisms mediated by FOXO family genes, which needs further investigation and confirmation. (Page 12, Line 313-317)
- Spelling and punctuation should be checked carefully.
Response: We are sorry for our careless mistakes. We agree with this suggestion, and we have carefully checked the manuscript and corrected the errors accordingly, which are highlighted in red in the manuscript.

Reviewer 2 Report
This manuscript presents the role of FOXO in various aspects of cancer biology, from a pan-cancer analysis point of view. The computational biology experiments are well done and solid. The figures are nice and informative. However, there are a few points that need to be addressed:
1 - RNA deconvolution for immune cell enrichment should be performed using CIBERSORT and xCell - the package selected is less well known.
2 - More discussion on why FOXO upregulation is usually protective in most cancers but deleterious in others.
3 - The SNV/CNV part of the paper is not clear. It should be more spelled what was done exactly and the presentation of the results must be improved.
4 - Many sentences lacking words or with structure making little sense, especially in the second half of the manuscript.
5 - The methylation analysis and its conclusions are unclear. More time to describe results and what exactly was done is needed.
Author Response
#Reviewer 2
1 - RNA deconvolution for immune cell enrichment should be performed using CIBERSORT and xCell - the package selected is less well known.
Response: Thank you so much for this valuable suggestion. As the reviewer says, this package “immunedeconv” is less well known than “CIBERSORT” or “xCell”. However, the advantage of this package is that it contains multiple algorithms to quantify tumor invasion, including “CIBERSORT”, “CIBERSORT-ABS”, “EPIC”, “ESTIMATE”, “MCPCOUNTER”, “QUANTISEQ”, “TIMER”, and “XCELL”. Besides, this package was published in Bioinformatics which confirmed its reliability (Sturm, G., Finotello, F., Petitprez, F., Zhang, J. D., Baumbach, J., Fridman, W. H., ..., List, M., Aneichyk, T. (2019). Comprehensive evaluation of transcriptome-based cell-type quantification methods for immuno-oncology. Bioinformatics, 35(14), i436-i445. https://doi.org/10.1093/bioinformatics/btz363). Moreover, we got consistent results after re-calculating with the package you mentioned, so we thought we could use this package to achieve the same kind of goal. Thank you again for your thoughtful comments.
2 - More discussion on why FOXO upregulation is usually protective in most cancers but deleterious in others.
Response: We totally agree with this suggestion. We have added the discussion as follows,
Despite overwhelming evidence supporting the FOXO family genes as tumor suppressors, some experts argued that the FOXO family genes could also be oncogenic. (Page 11, Line 267-269)
Our study found that the FOXO4 was a significantly protective factor in DSS, OS and PFS for THYM, MESO, LGG, and KIRC, while a highly risk factor in DFS, OS, and PFS for DLBC (Diffuse Large B-cell Lymphoma). The previous study consistently revealed that diffuse large B-cell lymphoma resistant to treatment exhibited enhanced FOXO4 expression and stem cell-like properties, reflecting a significant association between FOXO4 expression in DLBC and poor prognosis [23]. Collectively, these results suggest that the FOXO family genes are commonly differentially expressed in pan-cancer and have different prognostic values in various cancers. However, the underlying mechanisms need further elucidation. (Page 11, Line 273-281)
3 - The SNV/CNV part of the paper is not clear. It should be more spelled what was done exactly and the presentation of the results must be improved.
Response: We feel sorry that we did not provide enough explanation of the SNV/CNV part. We have added detailed explanation of the SNV/CNV part. Besides, we have improved the presentation of the results. The revisions are as follows,
SNV mainly refers to the variation of DNA sequence caused by the alteration of a single nucleotide at the genomic level, which is widely involved in tumor initiation, development, and metastasis. We further explored the variant landscape of FOXO family genes. The results were shown in Figure 2B-D. The total deleterious mutation percentage showed no mutation of the FOXO6 gene in pan-cancer. In contrast, the SNV mutation of the other three FOXO family genes frequently occurred in multiple tumors, especially in UCEC. Multiple kinds of variation were found, including frameshift deletion mutation, missense mutation, nonsense mutation, frameshift insertion mutation, inframe deletion, splice site, etc. A further finding was that FOXO1 and FOXO4 were the most frequently mutated (39%) among FOXO family genes in pan-cancer, and FOXO3 also had a high mutation frequency (34%). (Page 3, Line 115-125)
Except for SNV, CNV and DNA methylation are also associated with tumor progression. It is well known that abnormal CNV is one of the critical molecular mechanisms of tumor development. (Page 5, Line 135-137)
These findings implied that FOXO family genes might be the CNV and methylation drive genes. (Page 5, Line 148-149)
Figure 3. The CNV alteration and the methylation levels of FOXO family genes. (A) The CNV percentage of FOXO family in each tumor type. (B) The heterozygous and homozygous CNV profile of FOXO family in each tumor type, including the percentage of amplification and deletion. (C) Bubble plot of the correlations between CNV and mRNA expression of FOXO family in pan-cancer (FDR, false discovery rate). (D) Heatmap of the different methylation levels of FOXO family in pan-cancer (FC, fold change; *p < 0.05, **p < 0.01, ***p < 0.001, ****p < 0.0001). (E) Bar plot of the correlations between the methylation levels and mRNA expression of FOXO family in pan-cancer (FDR, false discovery rate).
4 - Many sentences lacking words or with structure making little sense, especially in the second half of the manuscript.
Response: Thank you for pointing out this problem. We really apologize for our mistakes, and we have carefully proofread the manuscript to minimize typographical, grammatical, and bibliographical errors, which are highlighted in red in the manuscript. For instance, “In addition, scientists have found that mutations, deletions, or amplifications of the FOXO family genes existed in various human cancers, indicating that the FOXO family genes might be attractive therapeutic targets. However, their roles in pan-cancer remain unclear, and the potential antitumor mechanisms need to be further elucidated.” (Page 11, Line 248-250)
5 - The methylation analysis and its conclusions are unclear. More time to describe results and what exactly was done is needed.
Response: We agree with this suggestion. As the reviewer says, more time to describe results and what exactly was done is needed. We have added detailed explanation of the methylation analysis to make its conclusion clearer. The revision is as follows,
DNA methylation is an epigenetic modification that leads to tumorigenesis and cancer progression by silencing tumor suppressor genes. We found that the DNA methylation levels of FOXO family genes were downregulated in most tumor tissues, indicating that they generally act as tumor suppressors (Figure 3D). Besides, Figure 3E showed that the expression of FOXO family genes was negatively associated with their promoter methylation levels in pan-cancer, especially in THYM. These findings implied that FOXO family genes might be the CNV and methylation drive genes. (Page 5, Line 143-149)

Round 2
Reviewer 1 Report
The authors significantly improved presentation of the manuscript.
However, the results concerning methylation of FOXO genes still remain unclear! After my comment, the authors changed the initial manuscript from “we found that most of these methylated genes were upregulated in tumor tissues” to "DNA methylation levels of FOXO family genes were downregulated in most tumor tissues” although this is NOT represented by the presented data in Figure 3D! Indeed, methylation levels are mostly increased which means that gene expression might be downregulated. There seems to be a misunderstanding between methylation level and gene expression! The authors should clearly clarify the difference between both terms! In this context, they should write “methylation levels of FOXO family genes are increased”, and gene expression is up- or downregulated!
Furthermore, the sentence “These findings implied that FOXO family genes might be the CNV and methylation drive genes.” (p.5, l.149) does not make sense to me.
Author Response
#Reviewer 1
The authors significantly improved presentation of the manuscript.
However, the results concerning methylation of FOXO genes still remain unclear! After my comment, the authors changed the initial manuscript from “we found that most of these methylated genes were upregulated in tumor tissues” to "DNA methylation levels of FOXO family genes were downregulated in most tumor tissues” although this is NOT represented by the presented data in Figure 3D! Indeed, methylation levels are mostly increased which means that gene expression might be downregulated. There seems to be a misunderstanding between methylation level and gene expression! The authors should clearly clarify the difference between both terms! In this context, they should write “methylation levels of FOXO family genes are increased”, and gene expression is up- or downregulated!
Response: We feel sorry that we had a problem with the use of color matching, which led to a mistake in our statement. We have corrected the palette of Figure 3D and adjusted the font size of figure legend. It is known that methylation levels are mostly increased which means that gene expression might be downregulated, and we have described this result in “Besides, Figure 3E showed that the expression of FOXO family genes was negatively associated with their promoter methylation levels in pan-cancer, especially in THYM”. Besides, we have corrected the statement as follows:
We found that the methylation levels of FOXO family genes are increased in most tumor tissues, indicating that they generally act as tumor suppressors (Figure 3D). (Page 5, Line 145-146)
Furthermore, the sentence “These findings implied that FOXO family genes might be the CNV and methylation drive genes.” (p.5, l.149) does not make sense to me
Response: We agree with your suggestion. We have removed Lines 149-150.

Reviewer 2 Report
The authors have done a good job of incorporating comments from myself and reviewer 1. The article is improved overall. There is still a major issue regarding methylation levels in Figure 3D.
I understand that methylation levels overall appear increased in general. However, authors state the opposite. It is unclear whether they were discussing about gene expression levels or not. The data in Figure 3D are clear - increased methylation. That's quite confusing.
Other more minor comments:
Lines 149-150 make no sense.
Line 265 hemangiomas is not clear. There are several types - infantile, congenital, cavernous... Please specify
Author Response
#Reviewer 2
The authors have done a good job of incorporating comments from myself and reviewer 1. The article is improved overall. There is still a major issue regarding methylation levels in Figure 3D.I understand that methylation levels overall appear increased in general. However, authors state the opposite. It is unclear whether they were discussing about gene expression levels or not. The data in Figure 3D are clear - increased methylation. That's quite confusing.
Response: We feel sorry that we had a problem with the use of color matching, which led to a mistake in our statement. We have corrected the palette of Figure 3D and adjusted the font size of figure legend. Besides, we have corrected the statement as follows:
We found that the methylation levels of FOXO family genes are increased in most tumor tissues, indicating that they generally act as tumor suppressors (Figure 3D). (Page 5, Line 145-146)
Other more minor comments:
- Lines 149-150 make no sense.
Response: We agree with your suggestion. We have removed Lines 149-150.
- Line 265 hemangiomas is not clear. There are several types - infantile, congenital, cavernous... Please specify
Response: Thank you for your advice. We have carefully read the corresponding reference and changed the sentences as follows:
Moreover, FOXO triple-knockout mice (FOXO1/3/4−/−) were prone to develop hemangiomas (endothelial cell hamartomas) and thymic lymphomas, confirming the FOXO proteins as genuine tumor suppressors [39]. (Page 11, Line 262-265)
